# FINE-TUNING LANGUAGE MODELS FOR FACTUALITY

**Katherine Tian**[*†]**, Eric Mitchell**[*†]**, Huaxiu Yao**[†§]**, Christopher D. Manning**[†]**, Chelsea Finn**[†]
[†]Stanford University  [§]UNC Chapel Hill
{kattian,eric.mitchell}@cs.stanford.edu

## ABSTRACT

The fluency and creativity of large pre-trained language models (LLMs) have led to their widespread use, sometimes even as a replacement for traditional search engines. Yet language models are prone to making convincing but factually inaccurate claims, often referred to as 'hallucinations.' These errors can inadvertently spread misinformation or harmfully perpetuate misconceptions. Further, manual fact-checking of model responses is a time-consuming process, making human factuality labels expensive to acquire. In this work, we fine-tune language models to be more factual, without human labeling and targeting more open-ended generation settings than past work. We leverage two key recent innovations in NLP to do so. First, several recent works have proposed methods for judging the factuality of open-ended text by measuring consistency with an external knowledge base or simply a large model's confidence scores. Second, the Direct Preference Optimization algorithm enables straightforward fine-tuning of language models on objectives other than supervised imitation, using a preference ranking over possible model responses. We show that learning from automatically generated factuality preference rankings, generated either through existing retrieval systems or our novel retrieval-free approach, significantly improves the factuality (percent of generated claims that are correct) of Llama-2 on held-out topics compared with RLHF or decoding strategies targeted at factuality. At 7B scale, **compared to Llama-2-Chat, we observe 53% and 50% reduction in factual error rate** when generating biographies and answering medical questions, respectively. A reference implementation can be found at https://github.com/kttian/llm_factuality_tuning.

## 1 INTRODUCTION

Recent developments in training large language models (LLMs), particularly methods that learn from rankings over responses such as reinforcement learning from human feedback (RLHF) (Christiano et al., 2017; Ziegler et al., 2020; Ouyang et al., 2022), have enabled the development of powerful, engaging dialogue agents. State-of-the-art LLMs are pre-trained on a vast amount of knowledge in large datasets (Touvron et al., 2023a;b) and further fine-tuned to apply this knowledge to follow diverse instructions or complete more specific tasks (Chung et al., 2022; Chen et al., 2021). However, despite these large language models' exposure to diverse datasets, they are prone to confidently generating incorrect claims. One recent study shows that GPT-3.5 (ChatGPT) produces false citations more often than not when asked to provide the authors of a given study (Agrawal et al., 2023). Nonetheless, other research has demonstrated that in simple question-answering settings, large language models *do* exhibit systematic markers of uncertainty that indicate their factually unreliable statements (Kadavath et al., 2022; Tian et al., 2023). These results suggest that language models internally represent the limits of their knowledge, leading us to ask: *Can language models be fine-tuned to leverage this internal awareness, to avoid making untrue statements in the first place?*

A key source of difficulty in training factual models comes in specifying an objective that adequately captures factuality. As an example, maximum likelihood, the most common objective for pre-training language models, does not always encourage factual predictions. Consider the question "Where was Yo-Yo Ma born?" A model that continues by near-deterministically producing the text "idk, probably Paris?" is nearly always correct, but receives extremely high loss if the pre-training

---

[*]Equal contribution.

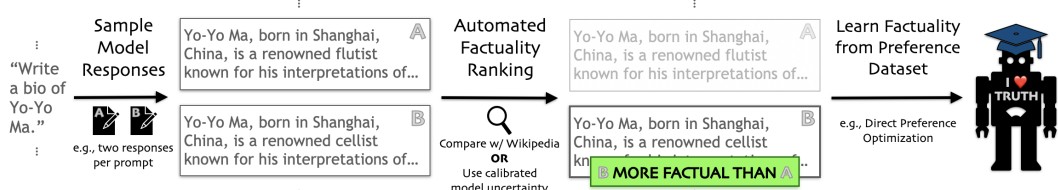

Figure 1: Our approach aims to improve the factuality of language models, specifically focusing on long-form generation (e.g. writing a biography). We develop two different approaches for estimating factuality of a passage (center), each of which allows us to generate a preference dataset (right). We then fine-tune the language model to optimize these factuality preferences (far right).

data contains any other response to the question. On the other hand, a model that hedges probability mass over many possible phrasings and many possible locations (including incorrect ones, like Antarctica) will likely receive much lower loss, as any response observed in the training data will be assigned at least *some* non-trivial probability. Because the pre-training objective may reward 'smearing' probability mass over many possible responses, language models may generate incorrect statements if they underfit the training data or if asked questions that require knowledge not contained in the pre-training data.

In principle, reinforcement learning-based objectives can avoid the failures of existing pre-training objectives through the appropriate choice of a reward function that penalizes factually incorrect statements. However, accurately computing such a reward function can be expensive. Obtaining human labels of factuality is time-consuming and costly; Min et al. (2023) report that professional fact-checkers took approximately 9 minutes to fact-check a single model-generated biography of a well-known individual; it cost about $2,000 to annotate 505 biographies.

In light of these challenges, we leverage recent advances in estimating truthfulness **without human intervention**: a) *reference-based* automated fact-checking methods that evaluate the extent to which an external knowledge base supports the claims in a piece of text (Min et al., 2023; Chern et al., 2023) and b) *reference-free* truthfulness evaluations that use a model's own confidence as a proxy for truthfulness, inspired by Kuhn et al. (2023). Using these truthfulness measures and a dataset of unlabeled prompts (e.g., "Write a biography of Yo-Yo Ma."), we sample pairs of completions from a pre-trained model and annotate them with a preference label denoting which has a lower rate of factual errors. Using the recently proposed Direct Preference Optimization (Rafailov et al., 2023) algorithm, we can stably and efficiently learn from such data. Ultimately, this pipeline enables us to fine-tune off-the-shelf language models to produce factual errors less often (with or without a reference knowledge base). See Figure 1 for an overview of our factuality tuning pipeline.

Our primary contribution is a straightforward approach to optimizing language models for factuality in long-form text generation without human annotation. We validate this approach on two benchmark datasets for evaluating factuality, targeted at generating biographies of popular figures and answering open-ended questions about medical conditions. We find that fine-tuning for factuality outperforms conventional RLHF and produces complementary benefits to LLM decoding strategies that aim to increase factuality. Further, we find qualitative differences in the result of learning from preference pairs scored with reference-based and reference-free truthfulness estimation. Overall, we find that learning factuality from automatically constructed preference pairs is a cost-effective way to increase model factuality without human intervention, reducing the error count for claims generated by Llama models by around 50% or more for biographies and over 25% for medical questions.

## 2 PRELIMINARIES

Our approach to fine-tuning directly for improved factuality uses the framework of reinforcement learning from preferences over candidate actions or responses. In this section, we provide an overview of reinforcement learning in the context of language models, as well as the specific algorithm we use for preference-based RL, direct preference optimization (Rafailov et al., 2023).

**Fine-tuning language models with reinforcement learning.** Reinforcement learning (RL) has proven to be an effective approach to fine-tuning language models to extract complex, useful behaviors from their pre-trained weights. In the context of RL, a language model policy $\pi_\theta$ (typically an

autoregressive Transformer) produces a conditional distribution $\pi_\theta(y \mid x)$ over responses $y$ given an input query $x$ (both $x$ and $y$ are text sequences). The goal of reinforcement learning is to maximize the average reward of outputs generated by the policy, where a reward function $r(x, y)$ assigns a scalar score to an input-output pair that determines its desirability. However, past works have observed that fine-tuning language models with an objective of unconstrained reward maximization can lead to *overoptimization* (Gao et al., 2022), that is, a policy that achieves high reward through exploitation of the idiosyncrasies of the reward function that are not aligned with the intended behavior. The most commonly-used objective in practice therefore combines reward maximization with a KL-divergence penalty between the language model and its initialization:

$$\max_{\pi_\theta} \mathbb{E}_{x \sim \mathcal{D}_p, y \sim \pi_\theta(y|x)} \left[ r(x, y) - \beta \log \frac{\pi_\theta(y \mid x)}{\pi_{\text{ref}}(y \mid x)} \right] \tag{1}$$

where $\mathcal{D}_p$ is some dataset of prompts, $\pi_{\text{ref}}$ is the reference model, usually the result of performing some supervised fine-tuning on a pre-trained model using demonstration data, and $\beta$ is a coefficient that controls the trade-off between reward and divergence (Ouyang et al., 2022; Bai et al., 2022; Stiennon et al., 2020). Optimizing this objective aligns the model with the reward function without deviating too far from the pre-trained reference model, reducing overoptimization. In practice, the most common algorithm used to optimize this objective for language models is proximal policy optimization (PPO; Schulman et al. (2017)), although some variants exist (Ramamurthy et al., 2022; Lu et al., 2022). However, these algorithms are quite complex to implement and tune (Zheng et al., 2023) and require online sampling during training, substantially increasing training time.

**RL from preferences with direct preference optimization (DPO).** Most large language models fine-tuned with Eq. 1 optimize a reward function that is *learned* from a dataset of preference rankings over possible model outputs. The DPO algorithm simplifies RL on language models for this special case (Rafailov et al., 2023), using a dataset of preference pairs $\mathcal{D} = \{x^{(i)}, y_w^{(i)}, y_l^{(i)}\}_{i=1}^N$ of prompts $x$ and candidate responses $y_w$ and $y_l$ (typically sampled from $\pi_{\text{ref}}$), where $y_w$ is preferred over $y_l$ (denoted $y_w \succ y_l$). The probability of observing a particular preference pair is assumed to follow a Bradley-Terry model (Bradley & Terry, 1952):

$$p(y_w \succ y_l) = \sigma(r(x, y_w) - r(x, y_l)) \tag{2}$$

where $\sigma$ is the sigmoid function and $r(x, y)$ is an unobserved reward or scoring function. Rafailov et al. (2023) show that the optimal policy $\pi^*$ for the problem in Eq. 1 can be found by optimizing a simple classification loss computed directly on the preference data:

$$\mathcal{L}_{\text{DPO}}(\pi_\theta; \pi_{\text{ref}}) = -\mathbb{E}_{(x, y_w, y_l) \sim \mathcal{D}} \left[ \log \sigma \left( \beta \log \frac{\pi_\theta(y_w \mid x)}{\pi_{\text{ref}}(y_w \mid x)} - \beta \log \frac{\pi_\theta(y_l \mid x)}{\pi_{\text{ref}}(y_l \mid x)} \right) \right] \tag{3}$$

DPO enables learning $\pi_\theta$ from a fixed dataset of preferences, without fitting an explicit reward function or sampling from the policy in the loop of training. These advantages make DPO an attractive choice for fine-tuning language models for objectives other than imitation. However, a challenge remains in constructing preference pairs that encourage greater factuality.

## 3 CONSTRUCTING PREFERENCES ENCOURAGING FACTUALITY IN LONG-FORM TEXT

While existing preference learning algorithms like DPO enable efficient, stable learning from objectives other than maximum likelihood, they require data in the form of preferences over possible responses to a prompt. In this section, we propose two classes of approaches to generating such preferences without human labeling effort. One class leverages existing methods to determine consistency with external reference texts as a measure of truthfulness; we propose another, which leverages calibrated model probabilities themselves as a proxy for truthfulness. For both approaches, we are computing an estimated **truthfulness score** over the claims in each generated response; the response with higher average truthfulness is taken as the preferred response. See Figure 2 for an overview of both procedures for truthfulness scoring. Note that truthfulness scoring is needed **only at training time**; at test time, we can sample from the model in the normal manner.

### 3.1 REFERENCE-BASED TRUTHFULNESS ESTIMATION

An intuitive approach to estimating truthfulness is by estimating the consistency of a given piece of text with a reliable reference text or knowledge base. Several recent works have introduced

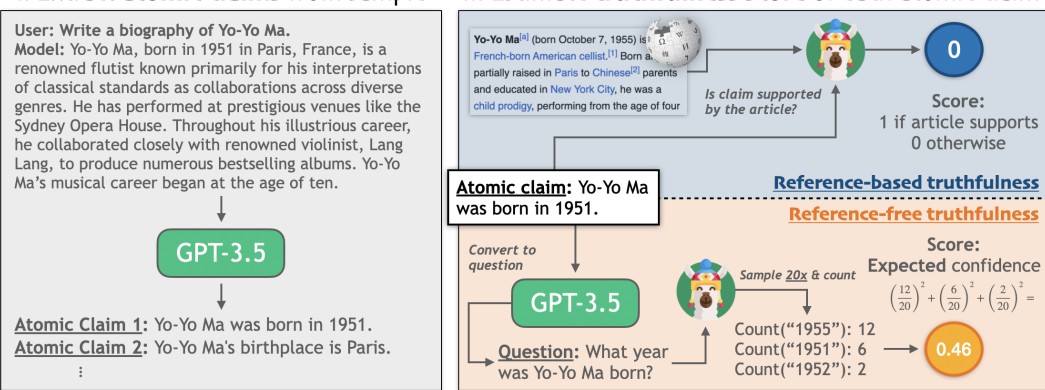

Figure 2: We estimate the factuality of a generation by first extracting claims (left) and then evaluating each claims' truthfulness (right). For the latter, we consider: a *reference-based* (top right) method that uses a fine-tuned Llama model to check if the fact is supported by Wikipedia (Min et al., 2023) and a *reference-free* (bottom right) method that uses the model's confidence in its most likely answer to estimate its truthfulness.

such evaluation criteria; for example, FactScore (Min et al., 2023) uses Wikipedia as reference knowledge, and FacTool (Chern et al., 2023) uses Google Search Results. These measures show high agreement with human judgments of factuality, making them attractive sources of truth for preference data construction. Due to the relatively consistent and high quality of Wikipedia articles, we elect to use FactScore as a representative method of reference-based truthfulness scoring.

To evaluate a piece of text, FactScore first extracts a list of the atomic claims present in the text using GPT-3.5.[1] For each atomic claim, a smaller, more efficient model such as a Llama-1-7b model (Touvron et al., 2023a) that has been fine-tuned for fact-checking is then used to perform natural language inference (MacCartney & Manning, 2008) to determine if a claim is supported by the reference text. The passage's truthfulness score is the fraction of the extracted atomic claims that are estimated to be supported by the reference text.

We note that reference-based truthfulness has the key limitation that it requires access to relevant, high-quality reference texts against which to measure consistency. Such a requirement may limit applicability to domains where ground truth documents are not known and accurate retrieval is difficult, such as in niche domains or less-structured tasks. Further, reference-based truthfulness estimation requires a reliable model to determine if an atomic claim is supported by the article. In light of these limitations, we propose a **reference-free** approach to estimating truthfulness of open-ended text, which avoids the need for retrieving external knowledge and checking consistency.

## 3.2 REFERENCE-FREE CONFIDENCE-BASED TRUTHFULNESS ESTIMATION

To eliminate the need for external knowledge, we leverage the fact that large language models are well-calibrated (Kadavath et al., 2022; Tian et al., 2023). That is, if a large language model assigns a fixed confidence $p$ to each claim in a set of claims, the fraction of these claims that is correct is $p$. In other words, in expectation over many claims, a perfectly-calibrated model's confidence in a claim corresponds to the probability it is correct. To use this notion of calibration, we interpret a model generation (e.g., a biography of Yo-Yo Ma) as a collection of claims, each resulting from a query to the model's knowledge (e.g., "When was Yo-Yo Ma born?" or "How many siblings does Yo-Yo Ma have?"). Our goal is to encourage the model to produce responses containing queries to its knowledge likely to lead to correct claims. Therefore, we parse a complete model generation into its constituent queries to the model's knowledge. For each query to the model's knowledge present in the generation, we can estimate the likelihood it will lead to a correct claim by simply estimating the average confidence of the model's answer to this query. If a model assigns probability 0.7 to '1955' and probability 0.3 to '1953' for the query "When was Yo-Yo Ma born?", then the probability this query will lead to a correct claim (again, in expectation over queries) is $0.7^2 + 0.3^2 = 0.58$. The model used for computing confidence scores essentially takes the place of the reference text

---

[1]https://platform.openai.com/docs/models/gpt-3-5

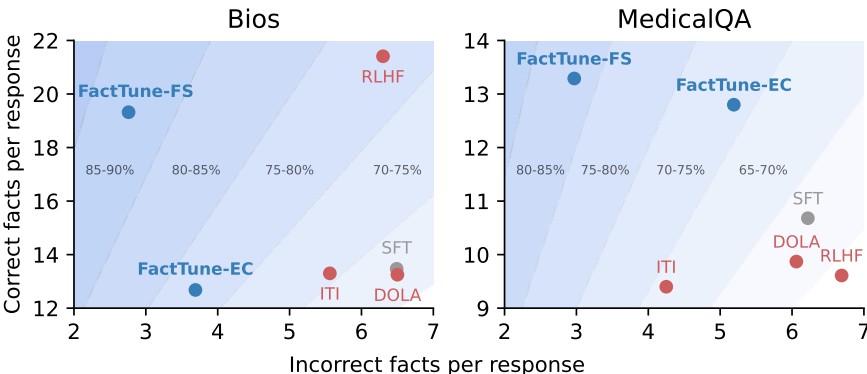

Figure 3: Factuality tuning using FactScore truthfulness scoring (FactTune-FS) produces by far the greatest improvement in factuality for the biography generation and medical question-answering problems. Factuality tuning with expected model confidence truthfulness scoring provides the next strongest performance, on average. For MedicalQA, only factuality tuning provides a **strict improvement** in factuality (more correct statements and fewer incorrect statements) compared to the SFT model.

datastore. We evaluate this *Expected Confidence* approach as well as a *Max Confidence* approach, which simply takes the max over the answer confidences for a given query (i.e., we assume the model produces answers greedily rather than sampling).

More concretely, we first extract atomic claims from the text using GPT-3.5. We then use GPT-3.5 to convert each claim to a query (question) testing knowledge of the particular fact. Careful rephrasing is necessary to ensure that the rephrased question is unambiguous; for example, the claim "Yo-Yo Ma plays the cello" should be converted to the question "What instrument does Yo-Yo Ma play?" rather than just "What does Yo-Yo Ma play?" as the latter question admits answers of the wrong type. If we were to use the second prompt, a model might assign 50% of its probability on "cello" and 50% of its probability on "basketball." However, the model's low confidence is caused by the ambiguity of the question, *not* low confidence in the instrument that Yo-Yo Ma plays. We detail the prompts used for question generation in Appendix A.2.

After each claim is converted to a minimally ambiguous question, we resample an answer 20 times from the base model (e.g. Llama-1-7b) that is fine-tuned to estimate the model's uncertainty over the answer. We use a few-shot prompt to encourage well-formed answers. We bin these answers by equivalence, using either heuristic string matching of the responses or using GPT-3.5 to assess if the answers are semantically equivalent, inspired by Kuhn et al. (2023). Our heuristic string match checks whether the words in the answer, excluding stop words, are the same. We compare these choices in Section 4.4. The score for each claim is either the expected or maximum confidence of the model's response; we finally average this score over all claims in a given model generation.

### 3.3 FACTUALITY TUNING: PUTTING IT ALL TOGETHER

Given a choice of truthfulness estimator, we can now construct a preference dataset for factuality tuning a given language model from a set of unlabeled prompts. First, we sample $n$ multiple candidate responses for each prompt from the model with simple temperature sampling with temperature 1.0 (using few-shot prompting for models that have not been fine-tuned). For each response, we then compute the truthfulness score with the chosen estimator (reference-based or reference-free). Finally, for all $\binom{n}{2}$ pairs of responses to each prompt, we simply choose the response with the higher truthfulness score as the preferred response. For a set of $m$ prompts, we ultimately generate $m\binom{n}{2}-k$ preference pairs, where $k$ is the number of pairs with equal scores. Finally, we fine-tune the model using the DPO pipeline, using all model responses as targets for the SFT stage.

### 4 EXPERIMENTS

Our experiments evaluate the extent to which factuality can be learned through preference-based reinforcement learning, using the fully automated preference-generation pipeline described in Section

| Dataset | Entities [train, val, test] | Prompts per Entity | Responses per Prompt | Example prompt |
|---|---|---|---|---|
| Biographies | 463 [288, 50, 125] | 1 | 10 | Write me a paragraph biography of Mary Wollstonecraft. |
| Medical QA | 295 [150, 45, 100] | 6 | 6 | What are the common symptoms of a stroke? |

Table 1: Dataset statistics and examples. In biographies, entities are individuals; in MedicalQA, entities are medical conditions. We include 6 questions for each entity in MedicalQA and adjust the number of responses per prompt to keep the total number of pairs in the two datasets roughly similar.

3. We call the model fine-tuned with our reference-based metric FactTune-FS and the model fine-tuned with our model confidence-based score, which is completely reference-free, FactTune-MC. For all of our experiments, samples for model confidence are taken from Llama-1-7b.

**Datasets.** We conduct our experiments on two tasks: generating biographies and medical question-answering. For biographies, we generated a dataset consisting of 463 diverse well-known individuals (288 train, 50 val, 125 test) with 10 short-paragraph biographies each. For medical question answering, we used a dataset of 295 diverse common medical conditions (150 train, 45 val, 100 test) with 6 questions about each condition and 6 short-paragraph answers per question. The test set just uses 1 question per condition. The prompts were generated with GPT-3.5, and the answers were sampled from Llama-1-7b using a few-shot prompt for each dataset. We found that our procedure consistently resulted in well-formed and informative responses, albeit with possible factual errors. Because FactScore uses retrieval against a given Wikipedia article, we generate data based on individuals and medical conditions that have Wikipedia pages. See Table 1 for the summary stats and examples from our datasets.

**Baselines.** We compare factuality tuning with inference-time intervention (Li et al., 2023, ITI) and decoding by contrasting layers (Chuang et al., 2023, DOLA), applied to the SFT model for each task. For ITI, we supervise the training of the linear probes with FactScore labels: we take batches of atomic facts extracted from the training samples and bias the models' activations from the incorrect to correct atomic facts to determine the direction of the intervention. In the case of Llama-2, we also compare against 'standard' RLHF with human preference labels (Touvron et al., 2023b).

**Evaluation.** To evaluate each generated response, we follow the FactScore procedure to extract the number of correct and incorrect facts. Then, to check that the model responses are still relevant and helpful after actuality fine-tuning, we also use GPT-3.5 to determine whether each fact is relevant to the question or not (using the prompt in Appendix A.2). For biographies, we observed that essentially 100% of facts were relevant to the individual, so we skip the relevance computation to save costs. For each dataset, we report the number of correct and relevant facts (# Correct), the number of inaccuracies (# Incorrect), and the proportion of correct relevant facts out of the total number of extracted facts (% Correct). Note that the total number of facts may vary between generations. We validate our evaluation metrics in Sec. A.1.

### 4.1 Fine-Tuning for Factuality Across Domains

In this section, we apply our methodology for learning factuality to Llama-1-7b and Llama-2-7b in multiple domains. We show the results in Table 2. Learning from reference-based factuality-scored pairs (FactTune-FS) consistently improves factual accuracy compared to RLHF models *and* decoding-based factuality baselines by at least 11% on biographies and 13% on medical question-answering. FactTune-FS reduces the number of factual errors and maintains no more than a slight decrease, if not increase, in the amount of correct information generated. Factuality tuning from model-confidence scores (FactTune-MC, FactTune-EC) also reduces error rate and improves the factuality of RLHF models on both datasets, without any external reference information.

### 4.2 Fine-tuning Chat Models for Factuality

Most widely used practical chatbots today are LMs trained with RLHF to follow diverse instructions in a way that is helpful to users. In this section, we investigate the ability of our human-free factuality tuning method to improve the factuality of RLHF chat models. Using Llama-2-7b-Chat, we find that fine-tuning an RLHF LM with both factuality and semantic entropy-based rewards can further improve its factuality without significantly decreasing the total number of facts, as shown in Table 3.

| Base Model | Method | Biographies | | | Medical QA | | |
|---|---|---|---|---|---|---|---|
| | | # Correct | # Incorrect | % Correct | # Correct | # Incorrect | % Correct |
| Llama-1 | ITI | 13.68 | 5.24 | 0.730 | 10.25 | 7.96 | 0.538 |
| | DOLA | 12.44 | 4.74 | 0.737 | 9.22 | 5.58 | 0.640 |
| | SFT | 13.54 | 6.54 | 0.696 | 9.96 | 6.86 | 0.600 |
| | FactTune-FS (Ours) | **14.51** | 3.74 | 0.812 | **12.60** | **4.18** | **0.746** |
| | FactTune-MC (Ours) | 9.74 | **2.42** | **0.819** | 11.51 | 5.56 | 0.668 |
| | FactTune-EC (Ours) | 10.84 | 3.28 | 0.790 | 11.52 | 6.56 | 0.641 |
| Llama-2 | ITI | 13.30 | 5.56 | 0.712 | 9.40 | 4.25 | 0.690 |
| | DOLA | 13.25 | 6.50 | 0.684 | 9.87 | 6.06 | 0.627 |
| | Chat | **21.41** | 6.30 | 0.774 | 9.61 | 6.69 | 0.619 |
| | SFT | 13.47 | 6.49 | 0.687 | 10.68 | 6.22 | 0.627 |
| | FactTune-FS (Ours) | 19.32 | **2.76** | **0.880** | **13.29** | **2.97** | **0.809** |
| | FactTune-MC (Ours) | 11.74 | 3.51 | 0.783 | 12.94 | 5.26 | 0.706 |
| | FactTune-EC (Ours) | 12.68 | 3.69 | 0.797 | 12.80 | 5.19 | 0.710 |

Table 2: Factuality tuning from reference-based factuality-scored pairs (FactTune-FS) improves factual accuracy compared to RLHF models and decoding-based factuality baselines, consistently reducing the number of errors *and* often increasing the number of correct facts generated. Factuality tuning from model confidence scored pairs (FactTune-MC, FactTune-EC) also outperforms RLHF models, providing a strong reference-free alternative for improving factuality and reducing error.

| Base Model | Method | Biographies | | | Medical QA | | |
|---|---|---|---|---|---|---|---|
| | | # Correct | # Incorrect | % Correct | # Correct | # Incorrect | % Correct |
| Llama-2-Chat | - | 21.41 | 6.30 | 0.774 | 9.61 | 6.69 | 0.619 |
| | DOLA | **22.25** | 5.81 | 0.793 | 11.45 | 6.74 | 0.624 |
| | FactTune-FS (Ours) | 20.02 | **4.38** | **0.821** | 11.94 | 6.21 | 0.667 |
| | FactTune-MC (Ours) | 19.12 | 4.97 | 0.795 | **12.61** | 7.21 | 0.627 |
| | FactTune-EC (Ours) | 18.77 | 5.13 | 0.784 | 11.51 | 6.40 | 0.639 |
| | OOD FactTune-FS (ours) | 21.06 | 5.45 | 0.796 | 11.56 | 6.66 | 0.635 |

Table 3: Factuality tuning a dialogue model (Llama-2-Chat) with FactScore, model confidence-based truthfulness estimation, and FactScore-based preferences from a different dataset (FactTune-FS, FactTune-MC, OOD FactTune-FS) further improves its factual accuracy more than a baseline method for factuality, DOLA.

In other words, **factuality tuning can be composed with RLHF to further improve the factuality of chat models.**

While our quantitative metrics demonstrate a clear increase in factual accuracy, we also investigate how factuality fine-tuning impacts other aspects of model performance and generalizes. Using GPT-4 as a judge, we find that **FactTune-MC and FactTune-EC can improve both factuality and fluency compared to the SFT model** (examples in Appendix Table 8). GPT-4 chooses FactTune-EC as more fluent than SFT on 80% of samples, FactTune-MC on 75% of samples, ITI on 57% of samples, FactTune-FS on 33% of samples, and DOLA on 16% of samples (n=100).

Lastly, we find that **fine-tuning for factuality generalizes across datasets**. Fine-tuning Llama-2-7b-Chat on biographies to evaluate on MedicalQA and vice versa (OOD FactTune-FS) improves the factuality more than RLHF (Table 3).

## 4.3 COMPLEMENTARY BENEFITS OF FACTUALITY TUNING AND DECODING-TIME FACTUALITY INTERVENTIONS

Besides fine-tuning for factuality, multiple existing works aim to improve LLM factuality through inference time interventions to either the decoding process or the model parameters themselves. We explore the possibility of applying both of these types of methods together, i.e., using factuality-boosting decoding methods on a model fine-tuned with our factuality tuning procedure. In Table 4 we present the results of stacking both approaches. We find that in most cases, DOLA can even further increase the accuracy of factuality fine-tuned models, with one exception for Llama-2 on the biography task. While not a comprehensive evaluation of combining methods for improving factuality, this result suggests that different approaches to enhancing factuality may operate through complementary mechanisms.

| Base Model | Method | Biographies | | | Medical QA | | |
|---|---|---|---|---|---|---|---|
| | | #Correct | #Incorrect | %Correct | #Correct | #Incorrect | %Correct |
| Llama-1 | FactTune-FS | 14.51 | 3.74 | 0.812 | 12.60 | 4.18 | 0.746 |
| | FactTune-FS + DOLA | 14.82 | 3.27 | 0.831 | 11.58 | 3.23 | 0.785 |
| Llama-2 | FactTune-FS | 19.32 | 2.76 | 0.880 | 13.29 | 2.97 | 0.809 |
| | FactTune-FS + DOLA | 18.82 | 2.81 | 0.873 | 13.13 | 2.67 | 0.830 |

Table 4: DOLA factuality decoding frequently composes with factuality fine-tuning, providing an increase in average correctness for the majority of combinations of model and dataset.

| Fact Ext. | Equiv | Metric | Biographies | | | Medical QA | | |
|---|---|---|---|---|---|---|---|---|
| | | | #Correct | #Incorrect | %Correct | #Correct | #Incorrect | %Correct |
| **Atomic** | Heuristic | Max Conf | 9.74 | 2.42 | 0.819 | 11.51 | 5.56 | 0.668 |
| | | Expected Conf | 10.84 | 3.28 | 0.790 | 11.52 | 6.56 | 0.641 |
| Entity | Heuristic | Max Conf | 12.22 | 4.74 | 0.742 | 10.32 | 6.94 | 0.605 |
| | | Expected Conf | 11.73 | 5.12 | 0.718 | 10.50 | 6.42 | 0.623 |

Table 5: On Llama-1, model confidence-based preference construction with atomic question extraction outperforms the version with entity extraction.

## 4.4 Impact of Design Decisions of Open-Ended Model Confidence Scoring

This section discusses the impacts of different design choices for the steps of our reference-free truthfulness score construction for factuality tuning: how to perform fact extraction and what confidence metric to use.

The first step is to extract the individual facts from the long-form response and re-sample each fact from the base model to assess the model's confidence in the fact. For the fact-extraction and resampling procedure, one approach (Atomic) is to convert each extracted atomic fact into a corresponding 'atomic question' with a few-shot prompt query to GPT-3.5, then sample answers to each question from the base LLM. Another approach (Entity) extracts entities from the response via `nltk` and re-samples the extracted entity in-line. Atomic question extraction has the potential to be more comprehensive and precise, while named entity extraction is a less expensive proxy that doesn't use closed models. In Table 5, we observe that atomic question extraction outperforms named entity extraction, although the difference in accuracy is smaller on Medical QA than on Biographies.

After re-sampling the fact, we study the choice of confidence metric between taking the model's confidence based on the most common sample (Max Conf) or the confidence of the fact from the original response (Expected Conf). To compute Max Conf for both atomic and entity extraction, we bin the samples into equivalence classes of distinct responses using a string matching heuristic described in Section 3.2 and take the proportion of samples in the largest bin. For computing Expected Confidence, we first perform the same answer binning procedure as for Max Confidence, resulting in $k$ bins and confidences $p_1, \ldots, p_k$, and take $EC = \sum_{i=1}^{k} p_i^2$. The results in Table 5 show that the performance of Max Conf versus Expected Conf varies but are quite similar.

## 5 Related Work

Many works have identified reducing factual errors (sometimes called 'hallucinations') as a key challenge for building more reliable language models (Lewis et al., 2020; Kadavath et al., 2022; Zhang et al., 2023), even for the most powerful language models (Bubeck et al., 2023). Other use of the term 'hallucination' refers to summarization or translation system outputs not supported by the reference text (Maynez et al., 2020; Zhang et al., 2020) even if they are factual (Cao et al., 2022). Other work uses 'hallucination' to describe vision-language models producing outputs not grounded in a visual input, e.g., a captioning system describing an object that doesn't exist in the image (Rohrbach et al., 2018). In our case, we focus on statements that are factually incorrect (or, inconsistent with a set of 'authoritative' texts, such as Wikipedia).

Several works describe methods for detecting likely factual errors through sensitivity to perturbations in the prompt (Xu et al., 2023), high diversity of responses under resampling (Kadavath et al., 2022; Mündler et al., 2023; Kuhn et al., 2023; Manakul et al., 2023), or inconsistency with exter-

nal knowledge sources (Min et al., 2023; Chern et al., 2023), or properties of internal activations (Azaria & Mitchell, 2023). Others go beyond detecting errors, correcting them after they have been generated (Peng et al., 2023; Gao et al., 2023; Dhuliawala et al., 2023). These approaches typically rely on retrieving relevant data from a trusted knowledge base and use another LLM to verify consistency; however, retrieval-based methods face key challenges, namely reliable resolution of conflicts between parametric and retrieved knowledge (Longpre et al., 2022; Chen et al., 2022) as well as maintaining improvements in factuality as model size increases (Mallen et al., 2023). Further, retrieval-based methods add significant system complexity; the most common open-source consumer language models thus use purely parametric models (Touvron et al., 2023a). The FactScore variant of our approach uses retrieval only during training, avoiding inference time complexity. In principle, any existing criterion could be used to generate preferences (see ; we aim to show that even choosing relatively simple criteria leads to substantial improvements in factuality.

Most similar to ours, some approaches attempt to prevent the generation of factual errors in the first place, using prompting strategies (Si et al., 2023) or perturbing the internal representations of the model (Chuang et al., 2023; Li et al., 2023). Unlike using a fixed heuristic for identifying an internal 'factuality' dimension, we optimize directly for the end goal of generating factual statements, which we find shows a greater improvement in factuality. Finally, while most past work has focused on short-form NLG tasks like short-form question-answering (Kadavath et al., 2022), we explore ways to measure model confidence over factual information in long-form, unstructured text and estimate truthfulness in a reference-free manner (i.e., don't require any external knowledge base or annotations).

## 6 CONCLUSION

In this paper, we show a practical, effective strategy to improve a language model's ability to generate factual content, specifically focusing on long-form generations. We develop and study two different approaches to estimating the truthfulness of long-form text and optimize for these criteria using preference-based learning. In addition to existing *reference-based* truthfulness estimators that leverage external knowledge to establish the truth of a particular statement, we introduce a novel *reference-free* procedure for estimating truthfulness that uses the language model's own uncertainty as an indication of factuality. Our experiments show that fine-tuning a language model with either criterion reliably reduces the number of incorrect facts (i.e. hallucinations) that the model generates. Reference-free approaches like the one we introduced provide a scalable self-supervision strategy to improve factuality, eliminating the need for a reference corpus of 'gold' texts.

The experimental results suggest a number of avenues for future work. First, because of the limited research and thus the limited benchmarks on the factuality of long-form language model generations, we proposed two new tasks to benchmark our approach. These tasks are representative of but do not fully cover the range of scenarios where we would hope to improve factuality. Furthermore, our experiments provide evidence for improving the factuality of dialogue models that are already fine-tuned with RLHF, but still leave open the question of how best to combine typical RLHF rewards and approaches with factuality rankings. Similarly, exploring additional ways to combine factuality tuning with existing methods for improving factuality, such as in our factuality tuning + DOLA experiment, may be a fruitful direction for future research. Further, future work might explore alternative approaches to constructing factuality preferences, such as using self-correction (Pan et al., 2023). Finally, we explore only 7B models in this work. Scaling up our factuality tuning recipe to larger models (and larger preference datasets) may reduce hallucinations even further.

## ACKNOWLEDGEMENTS

EM gratefully acknowledges funding from a Knight-Hennessy graduate fellowship and a Stanford Accelerator for Generative AI and Education grant. CF and CDM are CIFAR Fellows.

**Reproducibility Statement.** We explain the steps of our fine-tuning method in Section 3. In Section 4.1, we provide details on the dataset (dataset statistics, how it was generated, and examples), as well as how the evaluation is completed and how we implemented the baselines. In the experiment subsections and captions, we provide additional implementation or reporting details. In the appendix, we provide the exact GPT-3.5 prompts used for the extraction steps of our reference-free scoring method. A codebase with instructions for factuality scoring, training, and evaluation can be found here: https://github.com/kttian/llm_factuality_tuning.

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

| Dataset | Evaluation | SFT | FactTune-FS |
|---|---|---|---|
| Biographies | Human | 0.582 | 0.846 |
| Biographies | FactScore | 0.669 | 0.921 |
| MedQA | Human | 0.662 | 0.838 |
| MedQA | FactScore | 0.534 | 0.806 |

Table 6: To validate that our models do not suffer from extreme reward overoptimization, we conduct a human evaluation of the Llama-1-7b SFT and FactTune-FS models and find that an increase in FactScore also corresponds to a large increase in human-annotated accuracy. This study is computed on a subset of 25 generations per model.

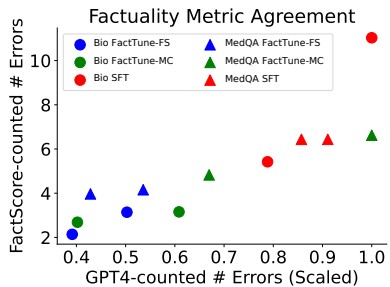

Figure 4: Average FactScore error counts and GPT-4 error counts are highly correlated, suggesting that the resulting models do not suffer from extreme reward overoptimization (Gao et al., 2022). We plot average FactScore error count v.s. average GPT-4 error count, scaling each dataset by the max GPT-4 error count in that dataset.

# A   APPENDIX

## A.1   VALIDATING METRICS FOR FACTUALITY

Our experiments primarily use counts of correct and incorrect facts computed by FactScore as the main evaluation metrics, as FactScore is automated and has been shown to exhibit good agreement with human fact-checkers (Min et al., 2023). Nonetheless, we aim to verify that our results are not specific or overfit to the FactScore criterion. In this section, we provide an evaluation with (1) human evaluators hired through Prolific.co[2] and (2) GPT-4.

To acquire human fact-checking results, we provide each human evaluator with a prompt, a generated response, and the title of the Wikipedia article they should use for fact-checking the response. We ask the human study participants to count the total number of facts and the number of incorrect facts in the response, and we divide these to obtain the human-rated accuracy. We provide the results in Table 6, where on average humans rated our FactTune-FS model for both datasets significantly higher than the SFT model.

Further, we ask GPT-4 to evaluate the factuality of a given response by counting the number of factual errors. We observe that the GPT-4 model ratings and FactScore model ratings are highly correlated, and GPT-4 provides another evaluation metric that demonstrates that FactTune-FS significantly reduces average error compared to the SFT models on both datasets (see Figure 4). Taken together, these results suggest that the improvements in factuality are not the result of exploitation of our evaluation protocol.

## A.2   PROMPTS

Table 7 contains the prompts used with GPT-3.5 to convert statements into questions for model confidence-based truthfulness estimation.

## A.3   SAMPLE MODEL GENERATIONS

See Table 8 for samples generated by several different models. After factuality tuning, the model does produce somewhat terser responses.

---

[2]Human evaluators were compensated at an estimated hourly rate of $16-18.

| | |
|---|---|
| Biography
Atomic Fact
to Question | I will provide a statement containing one atomic fact related to Hillary Clinton or people around her. Please rephrase the following statement into a specific question testing knowledge of the key fact in the statement. For example:
Statement: Hillary Clinton was born in 1947.
Question: In what year was Hillary Clinton born?
Statement: Hillary attended the Wellesley College.
Question: What college did Hillary Clinton attend?
Statement: She married Bill Clinton.
Question: Who did Hillary Clinton marry?
I will provide a statement containing one atomic fact related to LeBron James or people around him. Please rephrase the following statement into a specific question that testing knowledge of the key fact in the statement. For example:
Statement: LeBron James is a professional basketball player.
Question: What is LeBron James' profession?
Statement: He is one of the best in the NBA.
Question: Where does LeBron James rank among NBA players?
Statement: James was born in Akron.
Question: In what city was LeBron James born?
I will provide a statement containing one atomic fact related to [NAME] or people around [HIM/HER]. Please rephrase the following statement into a specific question testing knowledge of the key fact in the statement. For example:
Statement: [STATEMENT]
Question: |
| MedicalQA
Atomic Fact
to Question | I will provide a statement containing one atomic fact about the medical condition menopause. Please rephrase the following statement into a specific question testing knowledge of the key fact in the statement. For example:
Statement: Menopause is a time in a woman's life.
Question: Menopause is a time in whose life?
Statement: Menopause is the time when a woman no longer has menstrual periods.
Question: Menopause is the time when a woman no longer has what?
Statement: There is a decline in the ovarian hormone estrogen.
Question: During menopause there is a decline in what?
I will provide a statement containing one atomic fact about the medical condition breast cancer. Please rephrase the following statement into a specific question testing knowledge of the key fact in the statement. For example:
Statement: The signs and symptoms include a lump or thickening in or near the breast or underarm.
Question: Do the signs and symptoms of breast cancer include a lump or thickening in or near the breast or underarm?
Statement: The signs and symptoms include a change in the size or shape of the breast.
Question: Do the signs and symptoms of breast cancer include a change in the size or shape of the breast?
I will provide a statement containing one atomic fact about the medical condition varicose veins. Please rephrase the following statement into a specific question testing knowledge of the key fact in the statement. For example:
Statement: Varicose veins occur when the veins under the skin become enlarged.
Question: Varicose veins occur when what happens to the veins under the skin?
Statement: Veins in the legs lose their ability to efficiently circulate blood back to the heart.
Question: Varicose veins in the legs lose their ability to efficiently do what?
I will provide a statement containing one atomic fact about the medical condition [CONDITION]. Please rephrase the following statement into a specific question testing knowledge of the key fact in the statement. For example:
Statement: [STATEMENT]
Question: |

Table 7: GPT-3.5 prompts used to convert atomic facts into questions.

| | |
|---|---|
| SFT | Greta Gerwig is an American actress and filmmaker. She was born in and raised in Sacramento, California. Gerwig became interested in the film industry in 2002, when she attended the San Francisco Art Institute. She got her start in 2006, when she starred in the film 'The Off Hours'. Her other films include Frances Ha, Mistress America, and 20th Century Women. In 2018, Gerwig directed Lady Bird, and received wide acclaim, a Golden Globe and Tony nomination for directing, and Oscar and BAFTA nominations for the film. |
| FactTune-FS | Greta Gerwig is an American actress and filmmaker. Born in Sacramento, California in 1983, Gerwig started acting at a young age. She is best known for starring in films such as 'Frances Ha' and 'Mistress America.' Gerwig was nominated for an Oscar for Best Actress for her performance in 'Lady Bird.' Gerwig has written and directed a number of acclaimed independent films, such as 'Frances Ha' and 'Mistress America.' |
| FactTune-MC | Greta Gerwig is an American actress and filmmaker. She was born in 1983 in Sacramento, California, and raised in Stanford, California and New York City. Gerwig has appeared in such movies as Greenberg, Frances Ha, and Jackie. She also portrayed the title role in the 2013 film of Jackie Kennedy. Her directorial debut is the 2012 movie, Lady Bird, which was nominated for the Academy Award for Best Picture. |

Table 8: Samples for writing a biography of Greta Gerwig under the SFT, FactTune-FS, FactTune-MC Llama-1-7b models.

