# OpenReview forum: "Fine-Tuning Language Models for Factuality"
_ICLR.cc/2024/Conference — ICLR 2024 poster_

### Official Review · Reviewer_mN58 · 2023-10-19

**Soundness:** 3 good
**Presentation:** 4 excellent
**Contribution:** 3 good
**Rating:** 6
**Confidence:** 4

**Summary:**

This paper studies approaches to improve the factuality of language models (LMs) by fine-tuning the LMs using reinforcement learning (the specific algorithm is DPO), and the two main types of the reward function are: 1) reference-based FactScore (referred to as “DPO-FS”) and 2) reference-free uncertainty measure based on self samples (referred to as “DPO-MC”).

The authors conduct the experiments on two tasks: 1) free-form generation of biographies and 2) medical question answering. The datasets were crafted specifically for their experiments, hence resulting in the small size (e.g., biography train/test = 296/59 and medical QA train/test = 150/50 instances). The base LM is Llama1 and Llama2. The main experimental results show that both DPO-FS and DPO-MC generate responses with a higher “correct” percentage than baselines (SFT and inference-time methods such as ITI and DOLA). Also, DPO-FS and DPO-MC achieve a higher correct percentage than Llama-2-chat. Lastly, the authors perform a human evaluation to validate the findings previously evaluated using GPT3.5.

**Strengths:**

The paper shows that DPO can be applied to improve the factuality of LMs as shown by DPO-FS and DPO-MC achieving better factuality, and to the best of my knowledge, the factuality-based reward has not been investigated yet. The paper also investigates both reference-free and reference-based, and shows the effectiveness of both methods.

**Weaknesses:**

1. Although existing work may have not used a factuality-based reward, the results in this paper are mostly the expected observations (e.g., applying RL-based training improves target rewards). For example, (Lu et al., 2022) applied RL (PPO) with a reward based on an external metric to improve toxicity, repetition, etc.

2. The main findings (Tables 2, 3, 4, 5) are all based on GPT3.5 evaluation, and coupled with the fact that the test sets are small (e.g., 59 instances for biographies & 50 instances for medical QA), I’m not certain how reliable the results are. Also, there is not much information in Section 5.5 about human evaluation, e.g., inter-annotator agreement, or how many annotators were employed.

3. How does the DPO fine-tuned model perform on out-of-domain tasks? For example, when fine-tuning to improve factuality on biographies, does it also improve factuality on medical QA? And does its general performance change?
There is also a recent survey paper (Pan et al., 2023) about aligning LLMs for different aspects (including hallucination/factuality), and I think it would be useful for authors to incorporate additional relevant papers (i.e., those that apply RL to improve LMs)

4. Weak base LM: This work uses Llama-7B as the base model, and at this size, the model may not yet be highly capable of long-form generation / medical QA. Previous works such as (Manakul et al., 2023) and (Mundler et al., 2023) investigated LLM hallucination with much larger LLMs (e.g., GPT3.5/4). It would be interesting to see, for example, when using larger models (either open-source such as larger Llama / Falcon-180B or private ones such as GPT-4), if the model still makes as many factual errors (because if they don’t – due to the emergent ability when scaling up – fine-tuning may not be necessary or have little impact).

There is a recent survey paper (Pan et al., 2023) about aligning LLMs for different aspects (including hallucination/factuality), and I think it would be useful for authors to incorporate additional relevant papers (i.e., those that apply RL to improve LMs)

*References*
- (Lu et al., 2022) QUARK: Controllable Text Generation with Reinforced Unlearning
- (Manakul et al., 2023) SelfCheckGPT: Zero-Resource Black-Box Hallucination Detection for Generative Large Language Models
- (Mundler et al., 2023) Self-contradictory Hallucinations of Large Language Models: Evaluation, Detection and Mitigation
- (Pan et al., 2023) Automatically Correcting Large Language Models: Surveying the landscape of diverse self-correction strategies

**Questions:**

My questions are related to the points in the weaknesses section. I'm looking forward to seeing your responses to the weaknesses above, especially point number 3.

---

> ### Author Response · Authors · 2023-11-22
> **Response 1/2**
>
> We really appreciate your thoughtful assessment of our work! To address your concerns:
>
> **Regarding novelty.** While our work indeed leverages DPO, an existing algorithm, usage of model confidences as a proxy for a learning signal for RL is to our knowledge completely novel, and we show this technique is very useful for improving factuality. A priori, it is not obvious that LLMs contain a strong enough internal model of what is true and false that they would be able to learn to increase their factuality in a generalizable way. That is, the novelty of our findings is that **we can directly fine-tune language models to improve their factuality from tractably-sized datasets, without human labels or test-time retrieval/repeated sampling.** Further, we also note the call for papers lists societal considerations including "fairness, safety, privacy" as specifically relevant topics to ICLR; in light of the important reliability and safety risks of language model hallucinations, we feel that our work showing substantial mitigation of this problem is highly relevant and will be of strong interest to the ICLR community.
>
> **Re: evaluation metric.** To clarify our evaluation, we use FactScore to measure factuality, which has an error rate of <2% compared to human annotations [1]. GPT-3.5 is used only to parse the claims from the text, not to evaluate their correctness directly (this is done by comparing the claim with a wikipedia article). Our human study additionally supports the reliability of FactScore.
>
> [1] FActScore: Fine-grained Atomic Evaluation of Factual Precision in Long Form Text Generation. Min et al., 2023.
>
> **Re: evaluation set size.** We generated newer, larger test sets using the same methodology as the originals. On these larger held out sets of 300 medical questions and 300 biography prompts, we find similar results as in our original test sets:
>
> On 300 medical questions & Llama2:
> | Method | Correct | Total | % Correct |
> | --- | --- | --- | --- |
> | SFT | 9.16 | 15.9 | 0.582 |
> | FactTune-FS | 12.6 | 16.2 | 0.783 |
> | FactTune-MC | 11.3 | 16.9 | 0.671 |
> | RLHF | 8.68 | 16.2 | 0.559 |
>
> On 300 biographies & Llama2:
> | Method | Correct | Incorrect | % Correct |
> | --- | --- | --- | --- |
> | SFT | 12.3 | 7.10 | 0.648 |
> | FactTune-FS | 16.0 | 3.13 | 0.841 |
> | FactTune-MC | 11.5 | 3.40 | 0.783 |
> | RLHF | 20.6 | 6.81 | 0.746 |
>
> The relative performance of our methods remains similar – FactTune still surpasses SFT and RLHF.
>
> **Re: generalization across domains.** In a new experiment, we show that factuality tuning generalizes very well across data distributions. We evaluate the result of training on one data distribution (e.g, bios) and evaluating on a completely different one (e.g., medQA). **In summary, factuality tuning on bios and evaluating on medQA (or vice versa) increases factuality more than RLHF, indicating that factuality tuning makes a meaningfully general improvement to the model's factuality.**
>
> Eval on Bios
> | Method | Correct | Incorrect | % correct |
> | --- | --- | --- | --- |
> | RLHF | 19.0 | 6.41 | 0.748 |
> | FactTune-FS | 21.0 | 4.50 | 0.824 |
> | (OOD-med) FactTune-FS | 21.0 | 5.31 | 0.799 |
>
> Eval on MedQA
> | Method | Correct | Incorrect | % correct |
> | --- | --- | --- | --- |
> | RLHF | 9.63 | 5.50 | 0.636 |
> | FactTune-FS | 9.50 | 5.63 | 0.680 |
> | (OOD-bio) FactTune-FS | 8.81 | 5.12 | 0.658 |
>
> **Re: factuality's impact on general sample quality.** To check for decreased capabilities, we measure the fluency of each model (fine-tuned from Llama-7b on biographies) compared to SFT, using GPT-4 as a judge. We find that **our approach leaves fluency of generated text essentially unchanged.** Further, DoLA harms model sample fluency (often through increased repetition).
>
> - DOLA: Preferred 34%
> - FactTune-FS (ours): Preferred 50%
> - FactTune-MS (ours): Preferred 48%

---

> > ### Author Response · Authors · 2023-11-22
> > **Response 2/2**
> >
> > **Re: additional literature on RL.** Thank you for this suggestion! To summarize, Lu et al., 2022 discuss an alternative approach to RL for language models (though it requires drawing samples during the training procedure), but does not discuss factuality. Manakul et al and Mundler et al describe other approaches to deriving reference-free truthfulness scores, which are compatible with our framework for factuality tuning; in contrast, our work is focused on showing that optimizing with RL even simple approaches to truthfulness scoring can lead to substantial reductions in hallucination. We have added more discussion of alternative RL algorithms in the preliminaries, discussion of the compatibility of alternative truthfulness scores in the related work, and the possibility of alternative approaches to preference construction through self-correction in the future work.
> >
> > **Re: strength of the base LM.** We repeated our experiments for a larger 30B Llama model, finding that the model still hallucinates quite often and that factuality tuning still provides reduction in errors.
> >
> > - SFT:                11.4 correct, 6.48 incorrect, 0.638 accuracy
> > - FactTune-FS: 12.5 correct, 5.84 incorrect, 0.693 accuracy

---

> > > ### Comment · Reviewer_mN58 · 2023-11-23
> > >
> > > Thank you for your detailed response & further experimental results to validate the findings, especially the generalizability across domains. All the points I made in my review have been addressed by the authors (e.g., evaluation, generalizability, model choice), and I've raised my rating.

---

### Official Review · Reviewer_7K7x · 2023-10-28

**Soundness:** 3 good
**Presentation:** 3 good
**Contribution:** 2 fair
**Rating:** 6
**Confidence:** 4

**Summary:**

This paper proposes fine-tuning language models to improve their factuality. Specifically, one reference-based and one reference-free method are explored to estimate the truthfulness of different model responses, the scoring/preference of which are then used to fine-tune the LMs with direct preference optimization.

**Strengths:**

- The paper writing is of high quality and easy to follow

- The proposed method, regardless of reference-based or reference-free, shows improved factuality than the SFT baseline and the highest correct% among the compared methods.

- The paper provides analysis and ablations of different variants such as fine-tuning the pretrained/chat models and combining with inference-time decoding method.

**Weaknesses:**

- [major] I have some concerns about the evaluation
  - The test sets (50 and 59 examples in each domain, respectively) look very limited, making it bit hard to understand the actual improvement of model factuality.  How reliable are the results? Is 75% -> 81% a lot? I can't really answer these questions after reading the paper.
  - I noticed that the total number of claims are often different for different methods. Could generation style (e.g., length) contribute to the seemingly better/worse results? I wonder if the authors have considered such factors.
  - There is also no evidence indicting the improved factuality doesn't come at the expense of performance in other aspects. I understand the authors may not have enough labor/compute for a more comprehensive eval like GPT or Llama but LLMs, in my experience, can behave in mysterious ways when you over index on one specific objective.

- [minor] The method is somewhat straightforward, which is not necessarily a bad thing if the evaluation can show meaningful improvements (that are worth fine-tuning specifically for factuality) than methods that modify decoding only.

**Questions:**

- I'm a little confused why choosing the largest bin to measure truthfulness in the reference-free setting. Does that mean the atomic claim doesn't really matter ("Yo-Yo Ma was born in 1951" and "Yo-Yo Ma was born in 1955" would both be converted to "What year was Yo-Yo Ma born")? So as long as two responses make a claim on the same fact, regardless if it's correct or wrong, they will receive the same truthfulness score? If the hypothesis is "a language model’s confidence in a generated answer is highly correlated with the probability that the answer is correct", why not use the distribution to cross-check like in the reference-based setting?

---

> ### Author Response · Authors · 2023-11-22
> **Response 1/2**
>
> Thank you very much for your thoughtful feedback! To address your concerns:
>
> **Regarding evaluation set size.** We generated newer, larger test sets using the same methodology as the originals. On these larger held out sets of 300 medical questions and 300 biography prompts, we find similar results as in our original test sets:
>
> On 300 medical questions & Llama2:
> | Method | Correct | Total | % Correct |
> | --- | --- | --- | --- |
> | SFT | 9.16 | 15.9 | 0.582 |
> | FactTune-FS | 12.6 | 16.2 | 0.783 |
> | FactTune-MC | 11.3 | 16.9 | 0.671 |
> | RLHF | 8.68 | 16.2 | 0.559 |
>
> On 300 biographies & Llama2:
> | Method | Correct | Incorrect | % Correct |
> | --- | --- | --- | --- |
> | SFT | 12.3 | 7.10 | 0.648 |
> | FactTune-FS | 16.0 | 3.13 | 0.841 |
> | FactTune-MC | 11.5 | 3.40 | 0.783 |
> | RLHF | 20.6 | 6.81 | 0.746 |
>
> The relative performance of our methods remains similar – FactTune still surpasses SFT and RLHF.
>
>
> **Gaining intuition about the results.** Another way to gain intuition about the results is considering error rate: an improvement of 75% to 81% correctness corresponds to a reduction in error rate from 25% to 19%. In other words, **the fraction of a model's response that is factually incorrect has decreased by 24%.** Our Llama-2 experiments show even more substantial improvement of a **50% reduction in hallucinations for biography generation and answering medical questions** on the extended test set.
>
> **Impact of length on results.** To address concerns that length is contributing to performance, we note that FactTune-FS produces a strict improvement over typical supervised fine-tuning, unlike RLHF. This fact is clearer in the following newly-added Figure 3, which shows that **only factuality tuning enables strict factuality improvement, defined as a simultaneous increase in factual statements AND decrease in incorrect statements**: https://imgur.com/a/bhrxGZX.
>
> **Impact of factuality tuning on general model sample quality.** As one way to check for decreased capabilities, we measure the fluency of each model (fine-tuned from Llama-7b on biographies) compared to SFT, using GPT-4 as a judge. We find that, for example, DoLA harms model sample fluency (often through increased repetition), while our approach leaves fluency essentially unchanged.
>
> - DOLA: Preferred 34%
> - FactTune-FS (ours): Preferred 50%
> - FactTune-MS (ours): Preferred 48%
>
> **Regarding the more general point of reward overoptimization.** While our previous two points have shown that factuality tuning does not come through exploiting length of responses or compromising the fluency of model samples, we also show that we do not over-exploit the training data by training on one data distribution (e.g, bios) and evaluating on a completely different one (e.g., medQA). **In summary, factuality tuning on bios and evaluating on medQA (or vice versa) increases factuality more than RLHF, indicating that factuality tuning makes a meaningfully general improvement to the model's factuality.**
>
>
> Eval on Bios
> | Method | Correct | Incorrect | % correct |
> | --- | --- | --- | --- |
> | RLHF | 19.0 | 6.41 | 0.748 |
> | FactTune-FS | 21.0 | 4.50 | 0.824 |
> | (OOD-med) FactTune-FS | 21.0 | 5.31 | 0.799 |
>
> Eval on MedQA
> | Method | Correct | Incorrect | % correct |
> | --- | --- | --- | --- |
> | RLHF | 9.63 | 5.50 | 0.636 |
> | FactTune-FS | 9.50 | 5.63 | 0.680 |
> | (OOD-bio) FactTune-FS | 8.81 | 5.12 | 0.658 |
>
> **Regarding novelty.** While our work indeed leverages DPO, an existing algorithm, usage of model confidences as a proxy for a learning signal for RL is to our knowledge completely novel, and we show this technique is very useful for improving factuality. A priori, it is not obvious that LLMs contain a strong enough internal model of what is true and false that they would be able to learn to increase their factuality in a generalizable way. That is, the novelty of our findings is that **we can directly fine-tune language models to improve their factuality from tractably-sized datasets, without human labels or test-time retrieval/repeated sampling.**  Further, we also note the call for papers lists societal considerations including "fairness, safety, privacy" as specifically relevant topics to ICLR; in light of the important reliability and safety risks of language model hallucinations, we feel that our work showing substantial mitigation of this problem is highly relevant and will be of strong interest to the ICLR community.

---

> > ### Author Response · Authors · 2023-11-22
> > **Response 2/2**
> >
> > **Regarding question on reference-free preference construction.** The intuition behind choosing the confidence of the max bin, rather than the sampled bin, is to encourage elicitation of facts that the model is confident in (even if the model gets unlucky and samples a low-probability false completion in some samples). The confidence of the max bin is highly correlated with entropy over the sampling distribution (>0.95) but more interpretable.
> >
> > Additionally, we expect the confidence of the max bin & the confidence of the sampled bin to be correlated as the max frequency option is most likely to be chosen. After computing the confidence of the sampled bin we found a 0.57 correlation between the scores.
> >
> > We hope this helps answer your questions about our work!

---

### Official Review · Reviewer_Pzp4 · 2023-10-31

**Soundness:** 3 good
**Presentation:** 4 excellent
**Contribution:** 2 fair
**Rating:** 5
**Confidence:** 4

**Summary:**

Authors propose to finetune Llama models for factuality in long-form generation tasks using DPO on automatically constructed preference pairs. Authors explore 2 methods for generating preference ratings: 1) Reference-based: Extracts atomic facts using GPT-3.5 and then use Lama-1-7B-based NLI model to determine correctness of each atomic fact with respect to the reference. Percentage of correct atomic facts is used to compare the factual correctness of samples. 2) Reference-free: Extracts facts using GPT-3.5, then use GPT-3.5 to convert a fact to a question (uses few-shot prompting). Then, through sampling answers multiple times from the model, they estimate the model's uncertainty for the actual answer. The model's uncertainty is used to compare the factual correctness of samples.

They evaluate their approach on two tasks: biography generation and open-ended medical QA. To accommodate for the reference-based metrics, they generate data based on individuals (for biographies) and medical conditions that have Wikipedia pages.

Results show superior factual accuracy for DPO-based models on both tasks.

**Strengths:**

- New results to show the benefit of using automated feedback for improving LLMs, targeting factuality for long-form open-ended generation.

- Paper is well-written, experimental settings are well-defined, human evaluation is performed.

- Paper also shows that DPO-finetuning is complementary to decoding-time factuality improvement method (DOLA), (DPO + DOLA outperforms DPO)

**Weaknesses:**

- Idea itself is not novel, RLAIF has been consistently shown to be useful (here, authors used DPO instead of PPO). Though the application is new.

- Including more fine-tuning based baselines can help understand the role of automated metrics. E.g., directly using prompts to compare factuality of two outputs w.r.t. the wikipedia article instead of extracting atomic facts.

- Both DPO variants reduce number of correct facts on biography generation. This does not seem very surprising, given the optimized metric is the percentage of correct atomic facts. For example, a sample with 10 correct and 5 incorrect is preferred over 11 correct and 6 incorrect. Can this bias be removed from the fine-tuned model, maybe by changing the metric or comparing samples of similar lengths? Or is it the bias of evaluation metric?

**Questions:**

Check questions in the Weakness section.

- Between reference-free and reference-based metrics, there is a significant difference in the total number (correct + incorrect) of generated facts (almost 30% fewer) on biographies. What's the source of this bias, any possible explanations?

- Could you provide statistics on the number of tokens in wining vs losing samples in all cases (dataset/model/metric)?

---

> ### Author Response · Authors · 2023-11-22
>
> Thank you very much for your thoughtful feedback on our work!
>
> **Regarding novelty.** While our work indeed leverages DPO, an existing algorithm, usage of model confidences as a proxy for a learning signal for RL is to our knowledge completely novel, and we show this technique is very useful for improving factuality. A priori, it is not obvious that LLMs contain a strong enough internal model of what is true and false that they would be able to learn to increase their factuality in a generalizable way. That is, the novelty of our findings is that we can directly fine-tune language models to improve their factuality from tractably-sized datasets, without human labels or test-time retrieval/repeated sampling. Further, we also note the call for papers lists societal considerations including "fairness, safety, privacy" as specifically relevant topics to ICLR; in light of the important reliability and safety risks of language model hallucinations, we feel that our work showing substantial mitigation of this problem is highly relevant and will be of strong interest to the ICLR community.
>
> **Regarding other fine-tuning baselines.** We asked GPT-3.5 to provide factuality preference pairs over our responses with CoT, but found that these preference pairs agreed with the FactScore-baed preference pairs only 55% of the time, suggesting that naively prompting GPT-3.5 does not have strong enough factuality signal.
>
> **Regarding impact of length on results.** To address concerns that length is contributing to performance: While other methods for factuality and RLHF either improve number of correct facts or decrease the number of incorrect responses compared to supervised fine-tuning, we note that FactTune-FS instead is able to improve both. This fact is clearer in the following newly-added Figure 3, which shows that factuality tuning is the only method that enables strict factuality improvement, defined as a simultaneous increase in factual statements AND decrease in incorrect statements: https://imgur.com/a/bhrxGZX.
>
> In some applications, if generating an additional false statement is more harmful than not generating an additional correct fact, in the trade-off between longer responses and more accurate responses, may also prefer the slightly shorter but more accurate ones from a model like FactTune-MC. Viewing generations of varying lengths as a selective prediction mechanism on generating accurate facts, we may want a model to adaptively learn (such as from its own confidences) to improve accuracy by learning when to stop generating more facts.
>
> Lastly, here are some length statistics:
> Average # words/sample in train set: 100 words
> Fact Score:
> - Avg # words in winning vs losing samples: 96 vs 105 words
> - Avg difference in each pair: winning has 9.4 more words than losing
> Model Confidence:
> - Avg # words in winning vs losing samples: 96 vs 105 words
> - Avg difference in each pair: winning has 8.7 fewer words than losing
>
> Average # facts/sample in train set: 24
> Fact Score:
> - Avg # facts in winning vs losing samples: 22.6 vs 25.1
> - Avg diff (facts) in each pair: winning has 2.4 fewer facts than losing
> Model Confidence:
> - Avg # facts in winning vs losing samples: 23.0 vs 24.7
> - Avg diff (facts) in each pair: winning has 1.7 fewer facts than losing

---

### Official Review · Reviewer_2run · 2023-11-01

**Soundness:** 3 good
**Presentation:** 3 good
**Contribution:** 2 fair
**Rating:** 6
**Confidence:** 3

**Summary:**

In this paper, the authors construct a direct preference optimization (DPO) dataset for improving factuality using reference-based and reference-free truthfulness annotation techniques. Through the proposed method, they improve accuracy in two tasks (Biographies and Medical QA) without human factuality labels. The authors demonstrate that the proposed method (DPO-FS and DPO-MC) can be applied to Llama-2 and Llama2-Chat, and combined with a factuality-decoding approach (e.g., DOLA).

**Strengths:**

- Motivation is intuitive and easy to understand
- The proposed method improves the truthfulness of LLM without human factuality labels
- The proposed method can be augmented with existing orthogonal approaches for factuality

**Weaknesses:**

- Because the framework is simple and the method of scoring truthfulness and fine-tuning technique uses existing approaches, the proposed method appears to have limited contributions

**Questions:**

In Biographies and Medical QA tasks, are DPO-FS, DPO-MC, and SFT fine-tuned on the training set of each dataset?

---

> ### Author Response · Authors · 2023-11-22
>
> Thank you very much for your feedback on our work! To address your questions:
>
> **Regarding novelty.** While our work indeed leverages DPO, an existing algorithm, usage of model confidences as a proxy for a learning signal for RL is to our knowledge completely novel, and we show this technique is very useful for improving factuality. A priori, it is not obvious that LLMs contain a strong enough internal model of what is true and false that they would be able to learn to increase their factuality in a generalizable way. That is, the novelty of our findings is that **we can directly fine-tune language models to improve their factuality from tractably-sized datasets, without human labels or test-time retrieval/repeated sampling.** Further, we also note the call for papers lists societal considerations including "fairness, safety, privacy" as specifically relevant topics to ICLR; in light of the important reliability and safety risks of language model hallucinations, we feel that our work showing substantial mitigation of this problem is highly relevant and will be of strong interest to the ICLR community.
>
> **Train sets.** DPO-FS, DPO-MC, and SFT are all fine-tuned with the train set of each dataset.

---

> > ### Comment · Reviewer_2run · 2023-11-23
> >
> > Thank you for the comment. After reading the comment and additional results in the general response, I have raised my score.

---

### Author Response · Authors · 2023-11-22
**General response and new changes**

We appreciate the reviewers feedback on our work! We'd like to highlight several new results and clarifications that address key questions reviewers had. In summary:

1. Factuality tuning **is the only method that increases number of correct statements while decreasing number of incorrect statements** (see new Figure 3)
2. Factuality tuning **generalizes well across domains** (training on bios, evaluating on medical QA, and vice versa)
3. Factuality tuning **does not harm sample quality**, as judged by GPT-4-rated fluency win rates vs the SFT model
4. **On larger, newly-expanded test set, we find the results of our experiments are unchanged**
5. Our work is first to show that RL trained against factuality preference labels generated without human labor offers **substantial reduction in hallucination**, which we feel will be of strong interest to the ICLR community.

**Regarding generalization to new domains.** We show that factuality tuning does not overfit the training data by training on one data distribution (e.g, bios) and evaluating on a completely different one (e.g., medQA). In summary, factuality tuning on bios and evaluating on medQA (or vice versa) increases factuality more than RLHF, indicating that factuality tuning makes a meaningfully general improvement to the model's factuality.

Evaluation on Biographies

| Model                | Correct Facts | Incorrect Facts | Accuracy (%) |
|----------------------|---------------|-----------------|--------------|
| RLHF                 | 19.03         | 6.41            | 0.748        |
| FactTune-FS          | 21.0          | 4.5             | 0.824        |
| OOD-medQA FactTune-FS| 21.0          | 5.31            | 0.799        |

Evaluation on Medicine

| Model                | Correct Facts | Incorrect Facts | Accuracy (%) |
|----------------------|---------------|-----------------|--------------|
| RLHF                 | 9.63          | 5.50            | 0.636        |
| FactTune-FS          | 9.5           | 5.63            | 0.680        |
| OOD-bios FactTune-FS | 8.81          | 5.12            | 0.658        |

**Regarding factuality's impact on general model sample quality.** As one way to check for decreased general capabilities, we measure the fluency of each model (fine-tuned from Llama-7b on biographies) compared to SFT, using GPT-4 as a judge. We find that, for example, DoLA harms model sample fluency (often through increased repetition), while our approach leaves fluency essentially unchanged (i.e., the win rate is very close to 50%).

- DOLA: Preferred 34%
- FactTune-FS (ours): preferred 50%
- FactTune-MC (ours): preferred 48%

**Regarding evaluation set size.** We have generated newer, larger test sets using the same methodology as the originals. On these larger held out sets of 300 medical questions and 300 biography prompts, we find similar results as in our original test sets:

On 300 Medical Questions & Llama2

| Model        | Correct Facts | Incorrect Facts | Accuracy (%) |
|--------------|---------------|-----------------|--------------|
| SFT          | 9.16          | 6.74            | 58.2         |
| FactTune-FS  | 12.6          | 3.60            | 78.3         |
| FactTune-MC  | 11.3          | 5.60            | 67.1         |
| RLHF         | 8.68          | 7.52            | 55.9         |

On 300 Biographies & Llama2

| Model        | Correct Facts | Incorrect Facts | Accuracy (%) |
|--------------|---------------|-----------------|--------------|
| SFT          | 12.3          | 7.10            | 64.8         |
| FactTune-FS  | 16.0          | 3.13            | 84.1         |
| FactTune-MC  | 11.5          | 3.40            | 78.3         |
| RLHF         | 20.6          | 6.81            | 74.6         |


While the exact numbers change due to the varying difficulty of questions, the relative performance of our methods remains similar – FactTune still surpasses SFT and RLHF.

**Regarding novelty.** While our work indeed leverages DPO, an existing algorithm, usage of model confidences as a proxy for a learning signal for RL is to our knowledge completely novel, and we show this technique is very useful for improving factuality. A priori, it is not obvious that LLMs contain a strong enough internal model of what is true and false that they would be able to learn to increase their factuality in a generalizable way. That is, the novelty of our findings is that **we can directly fine-tune language models to improve their factuality from tractably-sized datasets, without human labels or test-time retrieval/repeated sampling.** Further, we also note the call for papers lists societal considerations including "fairness, safety, privacy" as specifically relevant topics to ICLR; in light of the important reliability and safety risks of language model hallucinations, **we feel that our work showing substantial mitigation of this problem is highly relevant and will be of strong interest to the ICLR community.**

---

### Public Comment · ~Zoher_Kachwala1 · 2024-04-05
**Reference-Free Confidence-Based Truthfulness Estimation**

Great paper, and in my opinion the simplicity of the approach is a huge encouragement. My question is regarding reference-free confidence-based truthfulness. The FactScore paper also uses non-parametric likelihood (section 4.1) as a way to automatically estimate FactScore. Any thoughts on the viability of that as another method for reference-free confidence based truthfulness? And intuition of how it would compare against the question-based tuning?

---

### Meta-Review · Area_Chair_TQWN · 2023-12-06

**Metareview:**

This paper presents a method for improving the truthfulness of language models via fine-tuning. Two sources of supervision are used: reference-based truthfulness, where claim decomposition + NLI is used to gauge whether text is correct or not, and reference-free truthfulness, which is based on an empirical estimate of the model's confidence via resampling and checking for semantic equivalence of answers. Paired preferences are used in DPO to fine-tune a model to generate more factual biographies (in the FActScore setting) and medical QA answers.

The reviewers liked the idea of this factuality-based reward and the fact that it does not rely on human factuality judgments.  The paper is very clearly written. The results show improvement over SFT and past inference-time methods like ITI and DOLA.

One critique is that the methods of scoring and then fine-tuning are simple and not novel. This is true, but given the importance of the problem of LLM factuality, a simple and effective method that outperforms past methods seems like a virtue.

Other critiques are raised about whether the results can be explained by a confounder like length.  The new Figure 3 addresses criticisms about changes in length being conflated with changes in factuality.  Furthermore, the authors have added several new experiments addressing domain generalization, overall sample quality, evaluation set size, and base LM size.

The new results address most of the criticisms, except for simplicity and novelty.

One additional reference the authors may wish to consider: Paul Roit et al. "Factually Consistent Summarization via Reinforcement Learning with Textual Entailment Feedback" https://arxiv.org/abs/2306.00186

**Justification For Why Not Higher Score:**

Lack of novelty, fairly simple method

**Justification For Why Not Lower Score:**

On balance this seems like a useful baseline on an important problem and a well-written paper. I think it's better than many other 5.75-6.00 in my batch. It has no strong advocate though, so I'm fine with it ultimately being rejected.

---

### Decision · Program_Chairs · 2024-01-16

Accept (poster)